# Anti-SARS-CoV-2 Vaccines and Monoclonal Antibodies Facing Viral Variants

**DOI:** 10.3390/v13061171

**Published:** 2021-06-18

**Authors:** Ahlam Chaqroun, Cédric Hartard, Evelyne Schvoerer

**Affiliations:** 1CNRS, LCPME, Université de Lorraine, 54100 Nancy, France; ahlam.chaqroun7@etu.univ-lorraine.fr (A.C.); c.hartard@chru-nancy.fr (C.H.); 2Laboratoire de Virologie, CHRU de Nancy Brabois, 54500 Vandoeuvre-lès-Nancy, France

**Keywords:** SARS-CoV-2, emergence, variant, mutations, immunity, vaccine, antibody

## Abstract

The severe acute respiratory syndrome coronavirus 2 (SARS-CoV-2) is genetically variable, allowing it to adapt to various hosts including humans. Indeed, SARS-CoV-2 has accumulated around two mutations per genome each month. The first relevant event in this context was the occurrence of the mutant D614G in the Spike gene. Moreover, several variants have emerged, including the well-characterized 20I/501Y.V1, 20H/501Y.V2, and 20J/501Y.V3 strains, in addition to those that have been detected within clusters, such as 19B/501Y or 20C/655Y in France. Mutants have also emerged in animals, including a variant transmitted to humans, namely, the Mink variant detected in Denmark. The emergence of these variants has affected the transmissibility of the virus (for example, 20I/501Y.V1, which was up to 82% more transmissible than other preexisting variants), its severity, and its ability to escape natural, adaptive, vaccine, and therapeutic immunity. In this respect, we review the literature on variants that have currently emerged, and their effect on vaccines and therapies, and, in particular, monoclonal antibodies (mAbs). The emergence of SARS-CoV-2 variants must be examined to allow effective preventive and curative control strategies to be developed.

## 1. Introduction

The emergence of SARS-CoV-2 in China and its rapid global spread is a source of concern [1]. This virus is responsible for the Coronavirus Disease 2019 (COVID-19) described for the first time in December 2019 in the city of Wuhan (Hubei province, China) [2].

SARS-CoV-2 is an enveloped virus [3] that has four major structural proteins, namely, Spike glycoprotein (S), envelope glycoprotein (E), membrane glycoprotein (M), and nucleocapsid protein (N), in addition to several non-structural proteins (NSPs). The S-glycoprotein has a major role in viral infection because it allows cell entry through binding of the virus to the angiotensin-converting enzyme 2 receptor (ACE2R), which is expressed mainly in the host’s respiratory tract and numerous other cells in the body. This glycoprotein is cleaved by proteases from the host cell into two subunits S1 and S2. S1 is responsible for determining the host range and cell tropism and includes a receptor-binding domain (RBD), which itself contains a receptor-binding motif (RBM) that interacts directly with ACE2R, whereas S2 mediates virus fusion in host cells [2]. Other receptors can also bind to S, such as integrin, through the arginine–glycine–aspartic acid motif (R403-D405) present in RBD [4].

The Spike protein has been identified as the immunodominant antigen of SARS-CoV-2. Indeed, neutralizing antibodies mainly target the RBD [5]. In addition to S protein, other proteins may have significant antigenic potential, such as N protein, M protein, and non-structural proteins (NSPs) [6]. The antigenic potential of these viral proteins allows triggering of a natural immune response involved in the recruitment of macrophages and monocytes, leading to the release of inflammatory cytokines in the infection site, in addition to an adaptive response mediated by CD4 + T lymphocytes (LTs) and cytotoxic LTs, allowing infection control. However, the immune system can be weakened (in elderly, immunocompromised, those with comorbidities, etc.,), and dysfunctional immune responses can occur and cause an immunopathology leading to serious disease, in particular, acute respiratory distress syndrome (ARDS) or sepsis symptoms. Thus, the severity of the disease is related not only to the viral infection but also to the immune host response with, among other factors, a low production of type 1 interferons playing a key role in the bad prognosis [7]. In addition to respiratory symptoms, SARS-CoV-2 infection induces vascular thrombosis and failure of multiple organs, including heart, liver, and kidney [8].

As a result, it is necessary to develop vaccine and therapeutic strategies based on host natural and adaptive immune responses to limit the progression of the disease, while considering the immunopathological side of certain responses, such as the phenomenon of antibody-dependent enhancement (ADE) [6,9].

Numerous vaccines and monoclonal antibodies against SARS-CoV-2 are in development or in clinical trials, and others are authorized for commercial use. However, vaccine and therapeutic efficacy has recently been threatened due to the emergence of several variants. Consequently, these variants may have a role in improving the transmissibility and severity of the disease, and the immune escape of the virus.

The first relevant event in this context was the occurrence of the D614G mutation in the Spike gene. Relatively early during the COVID-19 pandemic, in late January 2020, the D614G mutant was correlated with the viral fitness improvement [10]. 

Similarly, the variant of concern (VOC) 20I/501Y.V1, which emerged in Spain and was sequenced for the first time in December 2020 in the United Kingdom (UK), was associated with an important number of mutations (eight on Spike, four on ORF1ab, three on ORF8, and two on N). This variant was later detected in numerous countries, including the United States (US), Canada, and France. It carries the N501Y mutation in the receptor-binding domain (RBD), and other mutations including the 69/70 deletion, which lead to a conformational change in the Spike protein and the P681H near the site of S1/S2 furin cleavage [11]. 

This variant has been correlated with increased transmissibility and possibly a higher fatality rate, but until now has appeared to remain susceptible to vaccination. 20H/501Y.V2 and 20J/501Y.V3 variants were then detected in South Africa in December 2020 and Brazil in January 2021, respectively. They spread in Europe with increasing cases in the UK and France. They share mutations with the 20I/501Y.V1 variant, and particularly N501Y, but do not contain the 69/70 deletion. Some mutations could affect the activity of vaccines, leading to the generation of less effective post-vaccine antibodies against the mutated target Spike protein, such as the E484K substitution, which is involved in the vaccine immunity escape and is present in 20H/501Y.V2 and 20J/501Y.V3 variants [12,13].

Since January 2021, other variants have emerged in France and classified among variants of interest (VOIs), such as 19B/501Y detected in Henri-Mondor hospital (Ile-de-France) as part of a cluster. It is characterized by four key mutations on Spike: L18F on N-Terminal-Domain (NTD); L452R on RBD, which could be involved in antibody neutralization escape and viral fitness improvement; N501Y (in common with 20I/501Y.V1); and H655Y (in common with 20J/501Y.V3), which could be involved in the Spike modulation and immune escape [14]. Another example concerns the 20C/655Y variant that emerged in Brittany. It is characterized by nine mutations, and one deletion in the Spike, in particular H655Y, the 144 deletion (in common with 20I/501Y.V1), D215G (in common with 20H/501Y.V2), and V483A, near to the E484K residue in 20J/501Y.V3. These mutations may be involved in post-vaccine or post-infection immune escape or mAbs efficacy decrease [14].

In addition to the involvement of some SARS-CoV-2 mutations in immune escape, the emergence of mutations is responsible for the generation of heterogeneous SARS-CoV-2 strains in the same individual, which are known as viral quasi-species. This might impact tissue tropism. This observation was previously made for human immunodeficiency virus (HIV), known for its important diversity relating to the following factors: a high replication level (i.e., 10^10^ virions/patient/day); recombination of several HIV subtypes during cell coinfection; a reverse transcription responsible for approximately 10^−4^ substitutions/nucleotide/replication cycle; and host selection pressure. Subsequently, this variability has been shown to be responsible in HIV for tropism to the cerebral microglia and tissue macrophages [15,16,17]. 

The emergence of SARS-CoV-2 variants can, in part, be due to animal origins, such as minks infected with the virus in Denmark. Minks could be a potential reservoir of SARS-CoV-2 in addition to other animals that have shown a high sensitivity to the virus [7,13]. It was reported in November 2020 that one of the four mink farms in France was contaminated by SARS-CoV-2 [18].

Moreover, similar to the case of HIV, coronaviruses zoonotic transmissions are possible. Several simian immunodeficiency virus (SIV) transmissions have led to different HIV lineages through a genetic adaptation. Regarding coronaviruses, they were also able to cross the species barrier from bats to humans through a potential acquisition of the pangolin coronavirus RBD, which has a higher nucleotide similarity with the human SARS-CoV-2 RBD. Moreover, it has been reported that a chimeric virus constitution by recombination is possible in the case of cellular coinfection by two coronaviruses, and this is probably in the case of vole or rabbit infection, which carries two types of coronavirus (alpha and beta coronavirus). In terms of emergence, the hedgehog could also be a good candidate [18].

In this review, we present SARS-CoV-2 emerging variants, which are becoming a source of concern (i.e., variants of concern, VOCs) and others that should be evaluated (variants of interest, VOIs), in addition to others that are under surveillance (variants under surveillance, VUSs). We aimed to focus on and highlight the potential impact of viral genetic plasticity on tissue tropism, vaccine candidates, and therapy involving monoclonal antibodies.

## 2. Genetic Variability

In addition to the frequent recombination, mainly in ORF1a followed by S [19], SARS-CoV-2 is characterized by a mutation rate ranging from 1.19 to 3.3 × 10^−3^/site/year [20,21,22]. These mutations occur despite the exonuclease corrective activity (NSP14) and are associated with selection pressure induced by host immune response and vaccine/therapeutic immunity or tissue differentiation. Among these mutations, some are linked to the immune system, particularly when present in the RBD, such as E484K and S477N [23,24]. It has also been shown that mutations in RNA-dependent RNA polymerase (RdRp), particularly the P323 mutation in Nsp12 initially observed in the D614G variant, accelerate RdRp, thus affecting its fidelity and promoting mutations in the viral structural proteins [25]. 

These genetic modifications are at the origin of the emergence of several variants of concern (VOCs) and of interest (VOIs).

### 2.1. D614G Variant

The D614G mutation was first detected in late January in Germany and China. It is found in the glycoprotein Spike, which mediates attachment to the human ACE2R [26]. Originally, it was sporadic compared to the Wuhan reference strain, but has been a dominant strain since April 2020, following its transition into Europe, North America, Oceania, and Asia [10,26]. This mutation has provided an adaptive fitness advantage to the virus and has been associated with lower RT-PCR Cycle thresholds (Cts), suggesting higher viral loads in patients and higher viral titers as pseudotyped virions [10]. 

Based on Spike sequences available on April 06, 2020, the prevalence of the D614G variant has been reported to be significantly correlated with the COVID-19 fatality rate in different countries (Table 1) [27].

### 2.2. 20I/501Y.V1 Variant (Lineage B.1.1.7)

The 20I/501Y.V1 variant was sequenced for the first time at the end of December 2020 in the United Kingdom. It has been estimated to be up to 82% more transmissible than other pre-existing variants [11]. It is characterized by 17 non-synonymous mutations, including at least 3 of the 8 present in Spike that have a biological importance: (1) N501Y, which has a role in enhancing the binding affinity to ACE2R; (2) the P681H mutation, which is located near the furin cleavage site and absent in other group B coronaviruses, and is considered to be the key to SARS-CoV-2 pathogenesis because viral entry requires the processing of the S protein by cellular proteases, which are responsible for Spike cleavage at the S1/S2 site (the furin cleavage site); and (3) the 69/70 deletion, which is located in the N terminal domain (NTD) of Spike, and could promote the viral transmissibility and affect the performance of certain RT-PCR diagnostic tests, leading to a false negative result for the detection of the S gene (e.g., Thermo Fisher TaqPath COVID-19). However, as in the cases of most tests targeting other genes, the impact of this mutation on the SARS-CoV-2 diagnosis is not expected to be significant [11,31]. 

In addition to Spike mutations, the ORF8 Q27stop mutation truncates the ORF8 protein and thus allows other downstream mutations to accumulate [32]. ORF8 is involved in SARS-CoV-2 immune evasion [33]. This suggests a potential role of the Q27stop mutation in compensating for the reduction in vaccine or antibody activity, probably caused by certain mutations of the 20I/501Y.V1 variant. 

According to screening test results reported on April 08, 2021, the 20I/501Y.V1 variant predominates in France, with a rate of 81.9% (Figure 1). Moreover, it may be correlated with an increased risk of death and severity of the infection (Table 1) [34].

In January 2021, the British health authorities reported 11 cases that were infected with a 20I/501Y.V1 variant that had acquired the E484K mutation also present in the 20H/501Y.V2 and 20J/501Y.V3 variants. In April 08, 2021 the CNRS and French Public Health Agency declared there were nine cases in France infected with this variant [34]. The acquisition of the E484K mutation is a source of concern because of its involvement in vaccine and mAbs immune evasion (Figure 2).

### 2.3. 20H/501Y.V2 Variant (Lineage B.1.351)

The 20H/501Y.V2 variant appeared in South Africa in December 2020 independently of B.1.1.7. This variant shares the D614G and N501Y mutations with the 20I/501Y.V1 variant. It is characterized by three main mutations in the SARS-CoV-2 Spike protein: E484K could be involved in a structural modification of the Spike end, thus allowing a potential escape from antibody neutralization or vaccination targeting the Spike protein, and the K417N and N501Y mutations may be involved in enhancing Spike binding to ACE2R [12,35]. The 20H/501Y.V2 variant is highly transmissible, as observed by its rapid increase in prevalence in the Eastern and Western Cape provinces within just a few weeks [35]. No evidence of increased disease severity has yet been identified (Table 1) [35].

### 2.4. 20J/501Y.V3 Variant (Lineage P.1)

The 20J/501Y.V3 variant was first detected in Japanese travelers returning from the Brazilian state of Amazonas in January 2021. It presents mutations in common with the 20H/501Y.V2 variant located at the RBM of the S protein: N501Y and K417T, which may be involved in enhancing transmission, and E484K, which may be related to a slight improvement in receptor-binding affinity and immune escape (Table 1) [13].

According to screening test results reported on April 08, 2021, the 20H/501Y.V2 or 20J/501Y.V3 variants are stable in France, with a rate of 4.2% (Figure 1) [34].

### 2.5. Mink and CAL.20C Variants

Investigation undertaken in Denmark has reported that farmed minks (*Neovison vison*) are susceptible to SARS-CoV-2. The results of the complete sequencing genome of the virus isolated from this animal showed the presence of a new variant that subsequently appeared in humans (Mink variant). It has therefore been suggested that minks may constitute a reservoir of SARS-CoV-2 due to the possible human-to-animal and animal-to-human transmission. Since June 2020, 214 COVID-19 human cases have been identified in Denmark with SARS-CoV-2 variants associated with farmed minks, including 12 cases with a single variant, reported on 5 November 2020 [30].

This variant carries a nucleotide mutation at position 22920, leading to the Y453F substitution on the RBM of the Spike protein. This mutation has been associated with reduced sensitivity to neutralizing antibodies of sera from coronavirus recovered patients [36]. To date, the Mink variant has not been associated with clinical severity or an increased in the number of deaths (Table 1) [37]. This mutation was observed in another variant (CAL.20C) in the United States in southern California. It derives from the B line (B.1.429 and B.1.427) and carries three mutations on the Spike: S13I, W152C, and, most notably, L452R, located in the RBD and potentially involved in mAbs neutralization escape (Table 1) [38,39]. According to an 8 April 2021 public risk analysis of CNRS and Health, B.1.429 has been sporadically detected in France, whereas B.1.427 was only detected once, in Ile-de-France [34].

### 2.6. 19B/501Y Variant (Lineage A.27)

The 19B/501Y variant was detected in January 2021 in France in a cluster of four people, including three hospital professionals and one family contact case of one of them. Phylogenetic analysis of these four patients’ sequences revealed a variant belonging to the 19B clade. It is characterized by the presence of 18 substitutions, among which 7 or 8 are located on the Spike protein, including L18F in NTD and L452R in RBD, which are also present in 20J/501Y.V3, 20H/501Y.V2, and CAL.20C variants, respectively. These substitutions may be involved in Abs neutralization escape and may also confer an adaptive and replicative advantage to the virus. The variant also carries N501Y in common with the UK strain or N501T common with the Mink strain. This residue could be involved in improving the stability of the Spike-ACE2R complex. The H665Y mutation observed in 20J/501Y.V3 was also identified in this variant; it can modulate the stability, immunogenicity, and interaction of Spike with ACE2R. Like H665Y, the Q677H mutation is located near the furin cleavage site and may be involved in rapid virus replication, in addition to RBD-ACE2R binding [40,41,42,43,44]. 

The 19B/501Y variant has been detected among several clusters. All of these are a priori closed and affected educational and healthcare establishments. In some cases, more than sixty cases were reported, in particular, concerning two hospital clusters and eight intra-family clusters in Ile-de-France, three clusters in Pays de la Loire, one cluster in Brittany, and six clusters in Nouvelle Aquitaine [34].

In addition, community transmission of this variant has been suspected in at least two French departments (Seine-et-Marne and Dordogne), in addition to three probable reinfections. The association of this variant with the severity of the disease is still under evaluation [34]. It represented 1.8% of the interpretable sequences reported in France during the #4 COVID-19 Flash survey (2 March 2021) and 0.2% during the #5 Flash survey (17 March 2021), but was not detected during the #3 (16 February 2021) (Figure 1) [34].

### 2.7. 20C/655Y Variant (Lineage B.1.616)

The 20C/655Y variant (B.1.616) was first detected in January 2021 in Brittany, France, in the context of clusters, mainly in healthcare settings. It was preferentially detected in the lower respiratory tract. It is characterized by nine mutations, and one deletion in the Spike; in particular, these include H655Y, which is also present in the 20J/501Y.V3 and 19B/501Y variants; the Y144 deletion common with 20I/501Y.V1; the D215G mutation in common with the 20H/501Y.V2 variant; and the V483A close to the E484 residue mutated in the 20J/501Y.V3 variant (E484K), which may be involved in post-vaccine or post-infection immune escape or a decrease in the efficacy of monoclonal antibodies. On April 07, 2021, 25 cases were infected with 20C/655Y in France, mainly in Brittany. Most cases have been linked to hospital cluster transmission and a few cases have been linked to community transmission. However, its association with disease severity has not yet been determined [34].

### 2.8. 20C/477N and 20C/484K (Lineage B.1.526) 

The B.1.526 lineage was detected in New York at the end of November 2020. This lineage has increased rapidly and carries Spike mutations including: D253G reported as an antibody escape mutation against the N-terminal domain; D614G (in common with 20I/501Y.V1, 20H/501Y.V2, and 20J/501Y.V3); and A701V, in common with 20H/501Y.V2 [45]. 

There are two main branches of this line. One branch has the S477N mutation (8.6% of B.1.526 viruses), which is found near the binding site of several antibodies and implicated in the enhancement of Spike-ACE2R interaction. Two cases with S477N have been detected in Auvergne-Rhône-Alpes in France. The second branch carries E484K common to 20H/501Y.V2 and the 20J/501Y.V3, and is present in 74% of B.1.526 viruses [14,45]. 

### 2.9. 20C/484K.V3 Variant (Lineage B.1.525)

The 20C/484K.V3 variant was detected in the United States and Nigeria in December 2020, and sporadically in France (Auvergne-Rhône-Alpes, Center-Val-de-Loire, Grand-Est, Hauts-de-France, Ile-France, Normandy, New-Aquitaine, Pays-de-la-Loire); it corresponds to the B.1.525 line, and carries a mutation on the 677th amino acid of the Spike protein located near the S domain cleavage site necessary for viral entry (Table 1) [44]. 

### 2.10. 20B/484K (P.2 Lineage) 

The 20B/501Y variant was detected on April 2020 in Brazil. According to the 8 April 2021 updated risk analyses of CNRS and French Public Health Agency, it has been detected sporadically in France, with more than 70 cases located in Guyane. It carries E484K, D614G, and V1176F on Spike. Its association with disease severity has not yet been determined [34].

### 2.11. 20B/501Y Variant (P.3 Lineage) 

The 20B/501Y variant was detected on January 2021 in Philippines. It was also detected in other countries (Norway, Germany, United Kingdom, Japan, Australia, New Zealand), but has never been detected in France according to the 8 April 2021updated risk analyses. It carries Spike mutations including E484K, N501Y, D614G, P681H/R, and V1176F. The association of this variant with disease severity has not yet been determined [34].

### 2.12. 20A/214Ins Variant (B.1.214.2 Lineage)

The 20A/214Ins variant has been classified as a variant under surveillance (VUS). It was initially detected in Switzerland, then in the United Kingdom from November 2020, and recently in Belgium. It is characterized by an insertion of three amino acids (Ins214TDR) and four mutations in the Spike gene: Q414K, N450K, D614G, and T716I. Some of these mutations are likely to alter the Spike affinity to receptors with a potential impact on transmission and neutralization efficiency. It has been reported in the international virological database GISAID that 46 cases were detected in France, in Ile-de-France, Brittany, Auvergne-Rhône-Alpes, Hauts-de-France, Grand-Est, and Pays de la Loire [34].

### 2.13. 20A/484Q Variant (Lineage B.1.617)

The 20A/484Q variant was first isolated in October 2020 in Maharashtra state (Mumbai). It carries mutations in NSPs and on some ORF proteins, in addition to two key mutations on RBD: L452R common to the CAL.20C variant and E484Q close to the E484K observed in some VOCs. This lineage is still under investigation; its transmission, severity, and vaccine/antibody resistance are still unresolved. However, the presence of L452R could be involved in the decrease in antibody activity, and the E484Q could affect vaccine efficiency. The presence of this double mutant (E484Q–L452R) could also promote transmission [29]. According to a national survey, the first cases were reported in France at the end of April 2021. Previously, a group of 24 nursing students from India, positive to B.1.617, passed through an airport in Paris on 12 April 2021 [29].

## 3. Viral Quasi-Species and Tissue Tropism

The respiratory tract is the main target of the SARS-CoV-2, and is subject to a variable infection gradient associated with higher expression of ACE2R in the nasal cavity and decreasing expression in the lower respiratory tract [46]. However, viral tropism is not only specific to this pathway and several noble organs are also affected by the infection, in particular, kidneys, cardiovascular system, brain, liver, spleen, lymphatics ganglia, and digestive tract (Figure 3) [47,48,49].

This vulnerability could be linked to the ubiquitous expression of ACE2R, which is higher in kidneys (100-fold higher more than in the lungs), and occurs in cardiac and digestive tissues [47,49].

Other receptors and co-receptors could also be involved in viral entry, such as CD147 glycoproteins, which are ubiquitous and highly expressed in proximal tubule cells, and Neuropilin-1 (NRP1), which is expressed in the olfactory epithelium, and could facilitate infection with SARS-CoV-2 and provide a potential pathway for the viral entry in the nervous system [48]. 

The Spike interaction with several tissues’ receptors/co-receptors could be modulated by the viral genetic variability. This involves the development of viral heterogeneous populations, which correspond to quasi-species (Figure 3). Moreover, certain mutations observed in emergent variants have been associated with receptor-binding improvement on Spike (Figure 4), such as theS477N mutation, observed in the 20A.(EU2) and the 20C/477N variants, and the H655Y mutation observed in 20J/501Y.V3, 19B/501Y, or 20C/655Y variants [38,50].

This tissue tropism could promote the dissemination of the virus in the environment (wastewater) through viral excretion in stools in the case of digestive replication, thus leading to potential fecal–oral transmission [52]. Gastrointestinal symptoms (nausea, vomiting, and diarrhea) have been observed in 2% to 80% of COVID-19 patients. In addition, SARS-CoV-2 RNA has been detected in stools even in pauci-symptomatic individuals [53]. However, the presence of infectious SARS-CoV-2 has not yet been confirmed in human stool or wastewater samples; rather, it has only been observed in a sparsely reproducible manner. Nonetheless, infectious particles have been isolated from nasal samples of ferrets inoculated with SARS-COV-2 originating from human stools, thus supporting the hypothesis of excretion of infectious viruses in stools [54,55].

## 4. SARS-CoV-2 and Immunity 

### 4.1. Natural and Adaptive Immunity

SARS-CoV-2 infects host cells via binding to the ACE2R and transmembrane protease serine 2 (TMPRSS2), and enters the cell by endocytosis. Active viral replication and release of the virus lead to pyroptosis of the host cell and the release of damage-associated molecular patterns (DAMPs). These are recognized by neighboring epithelial cells, endothelial cells, and alveolar macrophages, thus triggering the production of pro-inflammatory cytokines and chemokines, which are the signature of SARS-CoV-2 infection [6,7], such as IL-1, IL-6, IP-10, MIP1α, MIP1β, and MCP1. These proteins attract other immune cells to the infection site (monocytes, macrophages, and T cells), promoting further inflammation with IFNγ release, which is involved in immunomodulation and produced by Th1. These T cells can also present antigenic epitopes (Figure 5a). Some analysis of samples of recovering COVID-19 patients has reported that these T cells trigger an immune response against the Spike S domain, membrane protein (M), and nucleocapsid (N), in addition to the nonstructural proteins such as NSP3, NSP4, and ORF8 [7,56].

This response leads to a stimulation of several immune cells (Figure 5b) [6,56,58].

In order of preference, CD8 + LTs can target Spike glycoprotein, membrane protein, NSP6, ORF8, and ORF3a. These cells act by cytotoxicity against virus-infected cells.

B-lymphocytes differentiate into plasma cells capable of producing neutralizing antibodies through the release of Th2 cells’ cytokines. The produced antibodies can target several SARS-CoV-2 immunogenic epitopes, including S1, N, ORF9b, and NSP5. Significant IgG responses targeting these epitopes have been detected in convalescent serum samples from recovered COVID-19 patients, whereas IgM and IgA responses have been weaker.

Alveolar macrophages recognize neutralized viruses by the Fc Fragment through the opsonization process, in addition to apoptotic cells, and eliminate them by phagocytosis.

In the case of a defective immune system, a dysfunctional response may take place. A high level of pyroptosis leads to an overproduction of pro-inflammatory cytokines, thus allowing the development of a cytokine storm that can generate multi-organ lesions and an acute respiratory distress syndrome (ARDS), due to an accumulation of immune cells in the infection site. This induces alveolar lesions and edemas. This immune response can also result from an aberrant Th2 response and the production of non-neutralizing antibodies that can promote viral infection through the antibody-dependent enhancement phenomenon. These antibodies can constitute an independent ACE2R viral entry path to infect macrophages, and lead to increased inflammation and tissue damage (Figure 5c) [7,57]. 

These dysfunctional immune responses, and the decreased antiviral control, such as low levels of interferon-gamma (IFNγ) or high levels of pro-inflammatory cytokines, could be linked to escape strategies mediated by viral proteins, through their interactions with essential innate antiviral immunity proteins. These proteins include NSP13 and ORF6, and may target the interferon (IFN) pathway or ORF9, which can indirectly interact with the mitochondrial antiviral signaling protein (MAVS). Moreover, ORF8 has shown a significant down-regulation of the expression of major histocompatibility complex class I (MHC-I) in various cell types, thus leading to the disruption of antigen presentation and affecting the destruction of the virus-infected cells by cytotoxic T lymphocytes (CTLs) [6]. 

### 4.2. Vaccinable and Therapeutic Immunity

Natural and adaptive immunological data underpin the development of SARS-CoV-2 vaccines and therapies. This is based on the use of antigenic epitopes, which trigger an immune response that limits the progression of the virus.

The Spike protein is the immuno-dominant antigenic target, and therefore the most widely used for the development of numerous vaccines that are in clinical trials or commercially available, such as mRNA-1273 (Moderna), AZD1222 (AstraZeneca ChAdOx1-S), BNT162 (Pfizer-BioTech mRNA), and Ad26.COV2.S (Johnson & Johnson), which are currently authorized for use in the United States and recommended in France [59]; in addition to NVX-CoV2373 (Novavax), which is currently in phase II in South Africa, and in phase III in the United Kingdom, USA, Mexico, and Puerto Rico. This is the only vaccine that has provided sterilizing immunity involved in the absence of RNA viral traces in the respiratory tract after experimental infections in macaques (Table 2) [59].

Other structural and non-structural proteins are also considered to be potential targets, such as the nucleocapsid protein (N), which is more conserved than the S protein and can trigger a humoral and cellular immune response against SARS-CoV-2. No specific vaccine for the N protein has yet been used in clinical trials. However, the ImmunityBio, Inc. & NantKwest Inc vaccine candidate, which is in phase I of a human clinical trial, is a human *Adenovirus* type 5 (hAd5) vector that encodes S and N (Table 2) [65]. 

It has been reported that this protein triggers significant immune responses mediated by CD4 + and CD8 + T lymphocytes in SARS-CoV-2-recovered patients. A subunit vaccine candidate (UB-612) with S1 and S2 from Spike, M, and N proteins is also under clinical trial (phase I + phase II/III) (Table 2) [65]. 

NSP3 also constitutes a potential target that is not currently in the clinical or preclinical phase. However, it could constitute a vaccine or therapeutic target due to its important sequence similarity to the other coronaviruses, in addition to its potential role in triggering immune responses mediated by CD4 + lymphocytes [65].

Another promising means to limit the progression of the virus is the use of monoclonal antibodies (mAbs), which are able to recognize SARS-CoV-2 antigenic epitopes, particularly immunodominant epitopes (RBD, RBM, and NTD), of the Spike protein. The mAbs present the advantage of a curative or preventive ability against infection in all individuals, including immunocompromised patients for whom vaccines do not work or are not sufficiently effective. mAbs enter the bloodstream immediately and therefore provide rapid protection for a few weeks or months. Vaccines take longer to be effective, but usually provide long-term protection. Hence, mAbs could supplement vaccines by limiting the progression of COVID-19. Several monoclonal antibodies against SARS-CoV-2 have been developed and are under evaluation. However, two targeting RBDs were approved in November 2020 by the Food and Drug Administration (FDA): Bamlanivimab (LY-CoV555 or LY3819253), and the combination of Casirivimab (REGN10933) + Imdevimab (REGN10987) (Table 2) [63,64].

## 5. SARS-CoV-2 Variants Facing Vaccination and Monoclonal Antibodies

In the current circumstances, in which vaccines and mAbs are under development or in clinical use, new emerging variants, such as 20I/501Y.V1, 20H/501Y.V2, 20J/501Y.V3 and others with mutations in the Spike or other viral proteins, present a risk of changes in antigenic epitopes leading to potential immune escape (Figure 2) [66].

### 5.1. Variants and Monoclonal Antibodies

To evaluate the neutralization activity of a set of monoclonal antibodies, Wang et al. created a SARS-CoV-2 pseudovirus including 20I/501Y.V1 and 20H/501.V2 mutations [67]. 

Their study showed that the antigenic impact of the N501Y mutation was limited to a few mAbs targeting the internal or external side of the RBD (the 910-30 antibody and the S309 antibody, respectively). The 144 deletion has also an impact on a few mAbs targeting the antigenic supersite in Spike NTD (the 5-24, 4-8, and 4A8 antibodies) (Figure 2) [67]. However, the 20H/501.V2 variant is of more concern because of its resistance to neutralization through the most potent mAbs, which target the superantigenic site in NTD and others targeting RBM (2-151, LYCoV555, C121, and REGN10933 antibodies) (Figure 2). This resistance is largely linked to the E484K mutation (Figure 2). The 20J/501Y.V3 variant, in addition to other variants containing the E484K mutation, may have similar results as the 20H/501.V2 variant (Figure 2) [67].

mAbs cocktails have been developed by a large number of academic and industry groups to overcome the emergence of resistance by variants of SARS-CoV-2 (Figure 2) [66]. This study also tested a combination of Regimens antibodies (REGN 10933 + REGN 10987), CoV2-2196 + CoV2-2130, and the combination LY-CoV 555 + CB6. The first two recombinations did not show any impact on the neutralization activity of three variants (20I/501Y.V1, D614G, and 20H/501Y.V2) (Figure 2), except that each of these potent combinations had a component that lost some neutralization activity, thus showing that one compensates for the loss of the activity of the other. However, the third combination was unable to neutralize the 20H/501Y.V2 variant [67].

Several studies agree with these results. Diamond et al. analyzed the impact of the neutralization of several mAbs on a panel of variants, including an authentic chimeric variant named Wash SA B.13 51, which contains the 20H/501Y.V2 Spike gene with other additional mutations (D80A, 242-244 del, R246I, K417N, E484K, N501Y, D614G and A701V), in addition to a panel of isogenic recombinant mutant variants (69–70 Del, K417N, E484K, N501Y, and/or D614G) [66]. 

They showed a decrease in neutralization activity by certain mAbs targeting RBM and NTD on the Spike protein, or even a loss of activity (half-maximal effective concentration (EC50) < 10,000 ng/mL) in the case of the Wash SA B.13 51 strain and the recombinant mutant with the E484K and N501Y mutations. However, an absence of neutralization reduction was observed in other monoclonal antibodies in the case of Wash SA-B.1351 compared to the E484K/N501Y/D614G mutants, which showed a reduction in neutralization. This is associated with the presence of a K417N compensatory mutation located at the extremity of RBM in Wash SA-B.1351 [66]. 

The study also tested the activity of several antibody combinations, in particular COV2-2196 with COV2-2130, which generally retained an inhibitory activity against all tested strains (Figure 2). The combination of S309 with S2E12 showed reduced activity (almost ten-fold) against the E484K/N501Y/D614G strain, although it retained good activity against Wash SA-B.1.351 strain, thus leading to the suggestion that additional mutations, such as K417N, compensate for the loss of neutralization activity because of the E484K mutation [66].

### 5.2. Variants and Vaccines

Studies have shown that B.1.351, P.1 lines, and B.1.1.7 with the E484K mutation are a source of concern in terms of vaccination, because of their potential resistance to sera from convalescents, immune sera from animals, and human sera from vaccinated patients (Pfizer-BioNTech, Astrazeneca, Johnson & Johnson, and Novavax vaccines) (Figure 2) [68].

Wang et al. also evaluated the activity of convalescent plasma, in addition to vaccinated sera (Moderna, or Pfizer-BioNTech), against the new emerging variants using the SARS-CoV-2 pseudovirus containing 20I/501Y.V1 and 20H/501Y.V2 mutations [67]. They observed no significant impact on the neutralizing activity of convalescent plasma (almost three times more resistant) or of the sera of the vaccinated (almost two times more resistant) against the 20I/501Y.V1 variant. The slight decrease in the neutralizing activity was linked to the S982A mutation located in the S2 domain of the Spike protein [67]. 

However, the 20H/501Y.V2 variant showed resistance to neutralization by convalescent plasma (almost 11- to 33-fold) and to sera from vaccinated persons (almost 6.5- to 8.6-fold). This resistance is linked to the E484K mutation (Figure 2), which allowed the authors to suggest its location in an immunodominant epitope recognized by vaccines [67].

The 20J/501Y.V3 variant and other variants containing the E484K mutation may have similar results (Figure 2) [67].

## 6. Conclusions and Perspectives

Based on the host’s natural and adaptive immunity, many vaccines and neutralizing antibodies are in development or in clinical trials. The aim is to target antigenic epitopes, which trigger the innate immune response, in addition to CD4+ LTs, cytotoxic LTs, and B-lymphocytes. However, the possible immunopathological factors in some cases should be taken into consideration to avoid any form of undesirable responses, such as an aberrant Th2 response or antibody responses linked to ADE (antibody-dependent enhancement). There are also additional challenges because, as in the case of HIV, the emergence of mutations requires that the antiviral strategy is continuously updated. To limit any kind of vaccine or therapeutic resistance of these variants and their probable vulnerability, a number of measures must be taken. These include updating and adapting vaccines. Several companies with authorized publicly available vaccines, or others in clinical trials (phase II/III), have declared that they are adapting their vaccines to new variants (Figure 2).

Novavax announced it has started to develop a new version of its vaccine including the S protein of emerging variants, in particular, those of the 20H/501Y.V2 and the 20J/501Y.V3 variants, which will be the subject of a clinical study starting in spring 2021. Pfizer-BioNTech are initiating a study on the safety and immunogenicity of a third dose of the Pfizer-BioNTech COVID-19 vaccine to assess the effect of a booster on immunity against emerging variants of circulating SARS-CoV-2. Similarly, Moderna declared that it has shipped doses of a new COVID-19 vaccine (mRNA-1273.351) to the National Institutes of Health (NIH), which are designed to provide better protection against the 20H/501Y.V2 variants. Finally, Oxford Astrazeneca aims to reduce the time required for the mass production of a version of its vaccine suitable for new SARS-CoV-2 variants.

More recently, the World Health Organization (WHO) recommended the use of the Janssen Covid-19 vaccine from Johnson & Johnson in France, due to its single-dose efficacy and its suitability against the three VOCs present in France.

The use of antibody cocktails targeting several antigenic epitopes is a promising strategy to limit the progression of the pandemic. Moreover, the combination of Regimens antibodies (REGN 10933 + REGN 10987), in addition to another combination comprising CoV2-2196 and CoV2-2130, have been able to maintain efficacy against the 20H/501Y.V2 and 20I/501Y.V1 variants. More antibody cocktails should be in preparation (Figure 2).

Targeting additional epitopes that are more conserved, such as low plasticity regions of B- and T-cells epitopes from Spike and Nucleocapsid, may prevent immune escape caused by mutations carried by circulating SARS-CoV-2 variants (Figure 2). As an example, some epitopes are present in highly conserved regions covering residues 150–200 and 250–315 from the start codon of the N gene [69].

Studies should be carried out on other emerging variants to identify the consequences for vaccines and therapeutic strategies to avoid facing new resistance. More transmissible variants are currently circulating in France, such as 19B/501Y and 20C/655Y, which carry multiple mutations involved in immune escape (e.g., H655Y), or the CAL.20C variant, which carries the E484K and the L452R mutations. Their susceptibility to vaccines and neutralization by monoclonal antibodies remain to be determined [68].

Finally, the vaccination of reservoir animals, such as minks, hedgehogs, and ferrets, could be adopted as a strategy. However, it remains difficult to apply, except in the cases of farmed animals.

Updates to vaccines against emerging variants, and the effectiveness of certain cocktails of neutralizing antibodies against the most concerning variants, provide hope for near-future success in the fight against the COVID-19 pandemic. Despite the good recognition of certain antigenic epitopes of the SARS-CoV-2 Spike, additional mutations in other genes, such as polymerases that allow better replication of the virus, can affect the efficiency of vaccines or therapies. This shows the importance of the sequencing and analysis of the whole genome to identify various regions that can contain compensatory mutations.

## Figures and Tables

**Figure 1 viruses-13-01171-f001:**
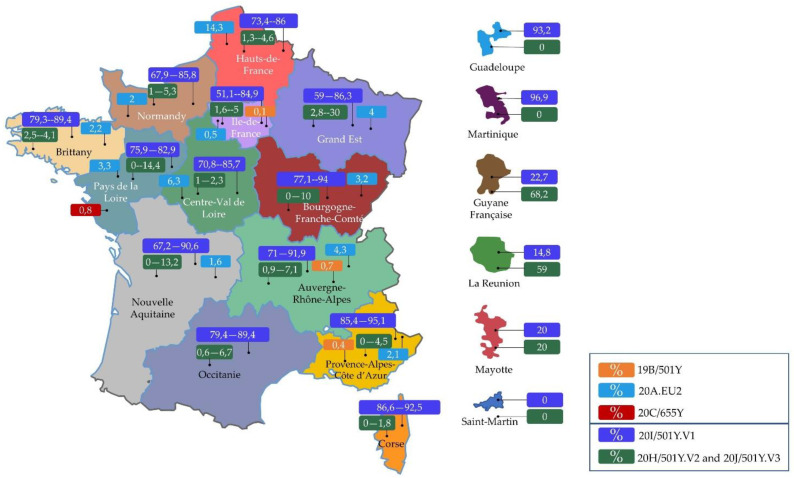
Percentage of three suspected variants of concern (20I/501Y.V1, 20H/501Y.V2, and 20J/501Y.V3) according to screening tests undertaken in France (7 April 2021), and percentage of 19B/501Y, 20C/655Y, and 20A.EU2 according to #5 Flash survey sequencing results (17 March 2021) [34]. (Designed by comersis).

**Figure 2 viruses-13-01171-f002:**
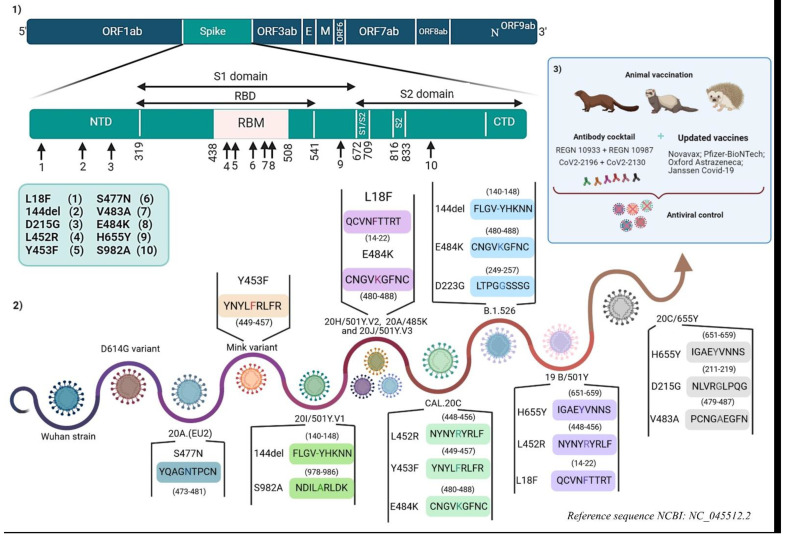
The history of SARS-CoV-2 and its variants subject to vaccine immunity and neutralization by monoclonal antibodies. (**1**) The localization of immune escape mutations in the SARS-CoV-2 genome. (**2**) Emerging variant mutations involved in vaccine immunity escape and/or monoclonal antibody neutralization escape. (**3**) Measures taken to control variants and the potential emergence of others. (Designed by biorender).

**Figure 3 viruses-13-01171-f003:**
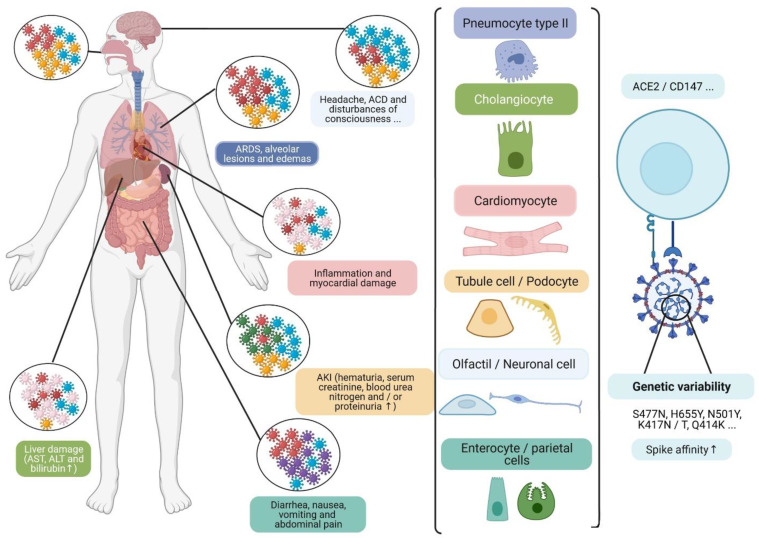
The development of viral quasi-species can modulate viral interaction with cell receptors in several tissues (type II pneumocytes, cholangiocytes, cardiomyocytes, tubular and podocyte cells, olfactory and neuronal cells, and enterocyte and parietal cells), particularly ubiquitous CD147 and ACE2R. This is, at least in part, due to the development of certain mutations, which may be associated with some clinical symptoms, and the elevation of certain biochemical markers in patients. Acute cerebrovascular disease (ACD); acute kidney injury (AKI); aspartate transaminase (AST); alanine transaminase (ALT); angiotensin-converting enzyme 2 receptor (ACE2R); differentiation cluster 147 (CD147) [48]. (Designed by biorender).

**Figure 4 viruses-13-01171-f004:**
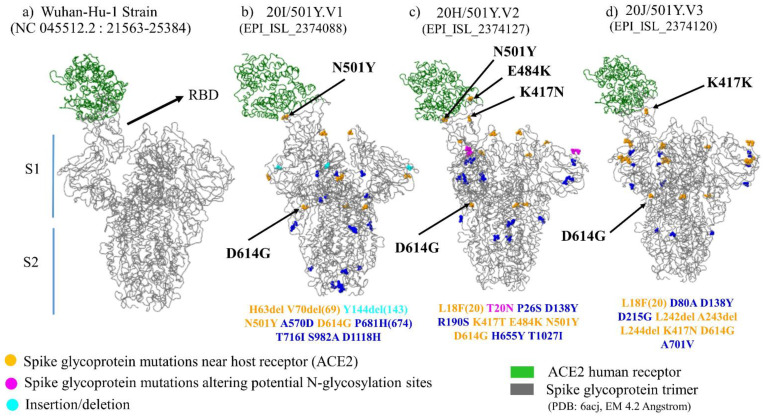
Tridimensional structure of Spike protein trimer in complex with host cell receptor ACE2. (**a**) 3D structure of Wuhan Strain Spike protein trimer; (**b**) 3D structure of Spike protein trimer of 20I/501Y.V1; (**c**) 3D structure of Spike protein trimer of 20H/501Y.V2; (**d**) 3D structure of Spike protein trimer of 20J/501Y.V3. S1 (Subunit 1); S2 (Subunit 2) [51].

**Figure 5 viruses-13-01171-f005:**
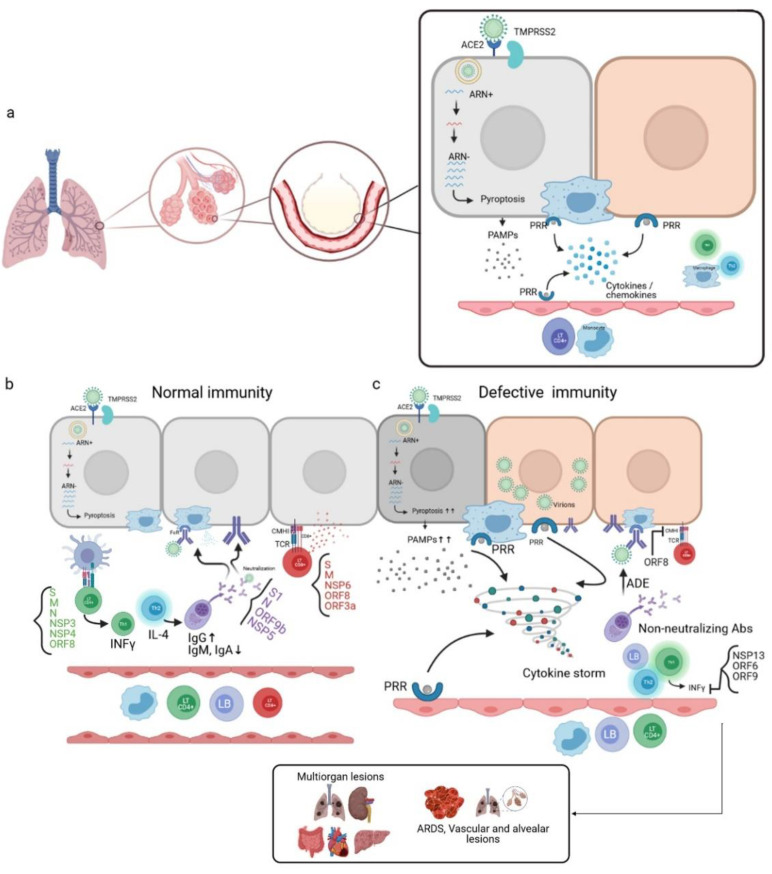
Mechanism of the immune response during SARS-CoV-2 infection. (**a**) SARS-CoV-2 penetrates the host cell by endocytosis through the Spike binding to the ACE2R present in alveolar epithelial cells. The active replication of the virus and the release of virions lead to pyroptosis, and, therefore, the release of pathogen-associated molecular patterns (PAMPs) recognized by pattern recognition receptors (PRRs) of neighboring alveolar epithelial cells, alveolar macrophages, and endothelial cells. This leads to the production of pro-inflammatory cytokines/chemokines, which in turn attract other immune cells to the infection site. (**b**) In the case of a normal immune system, CD4 + LTs recognize antigen-MHCII complexes of antigen-presenting cells (APCs). This induces the differentiation of CD4 + LTs into Th1, which is the producer of INFγ. CD4 + LTs can also differentiate into Th2, the producer of IL-4, which activates the LBs into plasma cells that produce neutralizing antibodies; these antibodies target several virus antigens and promote viral phagocytosis through the opsonization phenomenon. The CD8 + LTs recognize the antigens presented by the infected cells through the interaction of their TCRs with the CMHI of the infected cells, thus leading to cell death by cytotoxicity. (**c**) In the case of a defective immune system, a high level of pyroptosis induces a cytokine storm; non-neutralizing antibodies are produced, which are responsible for the antibody-dependent enhancement (ADE) phenomenon. In addition, inhibition of the production of INFγ occurs due to the involvement of certain viral proteins responsible for immune escape. This unbalanced immune response leads to multiorgan lesions, acute-respiratory-distress-syndrome (ARDS), and alveolar and vascular lesions [7,57]. (Designed by biorender).

**Table 1 viruses-13-01171-t001:** SARS-CoV-2 emerging variants of concern and variants of interest in France, and an example of a variant under investigation (according to CNRS and French Health Public Agency risk analyses updated on 08 April, 2021, and SFM databases updated on 29 April, 2021) [14,28,29].

Variant	Date/Country of the 1st Detection	Key Mutations	Location	Potential Role/Major Impact
D614G [10]	End of January 2020/Germany and China	D614G ^1^	CTD-S	Fitness/transmission advantage Correlation with lethality
Mink [30]	Denmark/ 7 November 2020	Y453F ^4^	RBM-S	Not associated to severity/death
		N501T ^2^		
20I/501Y.V1 (B.1.1.7) [14] (VOC)	United Kingdom/ End of December 2020	69/70 Del ^1,3^	NTD-S	up to 75% higher transmissibility Death (up to 82%)/severity increase Impact of S diagnosis kit
		144Y Del ^4^	NTD-S	
		N501Y ^2^	RBM-S	
		A570D	S1 region-Spike	
		D614G ^1^	CTD-S	
		P681H ^7^	Furin cleavage site	
		Q27stop ^5^	ORF8	
20I/484K/Q (B.1.1.7) [14] (VOC)	United Kingdom/ January 2021	E484K ^4^/Q	RBD-S	Potential immune escape for the 20I/484K/Q variant
20H/501Y.V2 (B.1.351) [14,28] (VOC)	South Africa/ December 2020	K417N ^2^	RBD-S	Highly transmissible (50%) Death (20%)/severity increase Post vaccine and infection immune escape
		E484K ^4^	RBD-S	
		N501Y ^2^/D614G ^1^	Spike	
		L18F ^4^	NTD-S	
20J/501Y.V3 (P.1) [14,28] (VOC)	Brazil/January 2021	E484K ^4^, N501Y ^2^, D614G ^1^	Spike	Transmissibility increase (up to 120%) Post vaccine and infection immune escape
		K417N/T ^2^	RBD-S	
		H655Y ^2,4,6^	Near the furin cleavage site	
20A/484K (B.1.525) [14,28] (VOI)	United States and Nigeria/ December 2020	69/70 Del ^1,3^, 144-145del ^4^, E484K ^4^, Q52R, A677H, F888L	Spike	Sporadic detection in France No evidence of health impact
CAL.20C [14,28] (VOI)	Southern California/May 2020	L452R ^4,6^	RBD-S	Slight increase in transmissibility (20%)
20C/477N and 20C/484K (B.1.526) [14] (VOI)	New York/February 2021	D253G ^4^	NTD-S	Some mutations may be involved in immune escape
		D614G ^1^, E484K ^4^	CTD-S	
19B/501Y(A27) [14] (VOI)	Henri Mondor Hospital-France/January 2021	L18F ^4^, L452R ^4,6^, N501Y ^2^, H655Y ^2,4,6^	Spike	Detection in the context of clusters Being evaluated Some mutations may be involved in immune escape
20C/655Y (B.1.616) [14] (VOI)	Brittany-France/January 2021	H66D	NTD-S	Detection in the context of clusters Being evaluated Some mutations may be involved in immune escape
		144-145del ^4^, H655Y ^2,4,6^	Spike	
		D215G ^4^	NTD	
		V483A ^4^	RBD-S	
20A/214Ins (B.1.214.2) [14] (VUS)	Swiss and recently in Belgium/November 2020	Ins214TDR	Spike	Being evaluated Some mutations may be involved in immune escape, transmissibility and ACE2R-S affinity increase
		Q414K ^2^, N450K ^4,2^, D614G ^1^		
		T716I ^2^		
20A.(EU2) [14,28]	Europe/Summer 2020	S477N ^2,4^	RBM-S	Majority of sequences in Europe in autumn 2020 Some mutations may be involved in immune escape, and ACE2R-S affinity increase
		D614G ^1^	CTD-S	
20B/484K (P.2) [14] (VOI)	Brazil/April 2021	E484K ^4^, D614G ^1^,	Spike	Being evaluated Detected sporadically in France
		V1176F ^2^	S2-Domain	
20B/501Y (P.3) [14,28] (VOI)	Philippines/January 2021	E484K ^4^, N501Y ^2^, D614G ^1^,	Spike	Being evaluated Not detected in France
		P681H ^7^,	Furin cleavage site	
		V1176F ^2^	S2-Domain	
20A/484Q (B.1.617) [29] (VOI)	India (Maharashtra state (Mumbai))/October 2020	E484Q, L452R ^4,6^	Spike	Under investigation

^1^ Promotion of transmissibility; ^2^ modulation of Spike and/or ACE2R interaction; ^3^ impact of RT-PCR diagnosis tests; ^4^ immune escape; ^5^ allowing accumulation of other mutations; ^6^ adaptive/replicative advantage, enhancing S1/S2 cleavage; ^7^ C Terminal Domain-Spike (CTD-S); N Terminal Domain-Spike (NTD-S); receptor Binding Motif-Spike (RBM-S); Receptor Binding Domain-Spike (RBD-S); variant of concern (VOC); variant of interest (VOI); variant under surveillance (VUS).

**Table 2 viruses-13-01171-t002:** Examples of SARS-CoV-2 vaccines (12 of 88 [60]) from the main types (from the 7 April 2021 infovac update) and two main monoclonal antibodies among those in clinical trials (from a 15 December 2020 review).

Vaccine Type	Vaccine/mAb and Sponsor Names	Description of the Vaccine/mAb	Type of Test	Number of Participant/Phase
Inactivated vaccine [61]	Coronavac (Sinovac and Butantan Institute)	Inactivated SARS-CoV-2 virus, with aluminum salts	phase I/II + phase III	144 (phase I); 600 (phase II) and 8870 (phase III)
	Inactivated SARS-CoV-2 Vaccine (Chinese Academy of Medical Sciences)	Inactivated SARS-CoV-2 virus	phase I/II	942
	BBIBP-CorV (Beijing Institute of Biological Products Co., Ltd. and Laboratorio Elea Phoenix S.A.)		phase I/II and phase III	448 (phase I); 1412 (phase II) and 3000 (phase III)
Subunit/purified vaccines [61]	NVX-CoV2373 (Novavax)	Nanoparticles containing Spike trimers with Matrix-M adjuvant	phase I/II, phase II in South Africa, and phase III (UK) + phase III (USA, Mexico and Puerto Rico)	1631; 2904; 15,000 and 30,000
	UB-612 (United Biomedical, COVAXX)	S1 and S2 subunits of the Spike protein and M and N proteins of SARS-CoV-2	phase I + phase II/III	60 and 7320
	SCB-2019 (Clover Biopharmaceuticals Australia)	SARS-CoV-2 Spike protein subunit in trimer form, with or without adjuvants (adjuvant 1: AS03/adjuvant 2: oligonucleotide “CpG 1018” + aluminum salts)	phase I + phase II/III	150 and 34,000
Vector vaccines [61]	AZD1222 (ChAdOx1 nCoV-19) (University of Oxford and Astra Zeneca)	Chimpanzee *Adenovirus* expressing the SARS-CoV-2 Spike protein	phase I/II + phaseIIb/III + phase III	1090; 10,260 and 40,050
	GRAd-COV2 (ReiThera Srl)	Inactivated gorilla *Adenovirus*, expressing the SARS-CoV-2 Spike protein	phase I	90
	Ad26.COV2.S (Johnson & Johnson)	Inactivated human *Adenovirus* type 26 expressing the SARS-CoV-2 Spike	phase I/II + phase III	1045 and 60,000
Genetic vaccines (DNA, RNA) [61]	BNT162 (Pfizer-BioNTech)	mRNA encoding the SARS-CoV-2 Spike protein, encapsulated in a lipid nanoparticle	phase I/II+ phase I/II + phase III	196; 7600 and 30,000
	mRNA-1273 (Moderna)	mRNA encoding the SARS-CoV-2 Spike	phase I + phase II + III	105,600 and 30,000
	AG0301-COVID19 (AnGes, Inc., Japan Agency for Medical Research and Development)	Spike encoding DNA plasmid	phase I/II	30
Monoclonal antibodies [62]	Bamlanivimab (LY-CoV555)/(Eli Lilly and Company) [63]	recombinant neutralizing human IgG1κ	Authorized by US Food and Drug Administration (FDA) in November 2020	465 (phase II)
	Casirivimab (REGN10933) + Imdevimab (REGN10987) [64]	Cocktail of Spike neutralization antibodies		799 (Phase I-III)

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
