# Peer review of "Anti-SARS-CoV-2 Vaccines and Monoclonal Antibodies Facing Viral Variants"

_viruses, 2021, doi:10.3390/v13061171_

Round 1

Reviewer 1 Report

this is a well written and timely review of the SARS Coronavirus 2 mutants and their role in viral transmission and immune response.  I really have only one criticism.  A schematic describing the organization of the genome and the location of the significant mutants is not shown until Figure 4.  This is really too late since it requires the reader to try to figure out what is happening.  What would be really nice would be if the authors could show a "three-dimensional" structure of the spike protein with the critical regions indicated and at least the first well known mutations shown.  This would allow the reader to put into perspective how the mutations might impact the structure of the spike protein and its interactions with receptors.  

Author Response

Dear reviewer,

Thank you for your advices. Please find attached the new version of the manuscript, including propositions of the 3 reviewers, as well as the 3-dimensional structure you have suggested.

Best regards,

Reviewer 2 Report

Chaqroun et al. present a review of the literature on the arms race between the causative agent of COVID-19 and the ongoing pandemic: SARS-CoV-2 and treatments developed to combat COVID-19: vaccines and antibodies. The article refers to developments on both sides, scientific data published by academic labs and pharma companies, and presents an argument that viral variants are evolving and will continue to evolve if the vaccines and antibodies are not updated to match circulating strains. This has been previously proposed for other evolving viruses, such as Influenza, where a strain update is needed every season. The article, however, suffers from lack of precision, poor fluency and needs a thorough review. The language is also very confusing and misleading. I have pointed out some of these points below, and would recommend that the authors take the time to work on a revised version of the article.

Title: Grammar needs checking. I speak English as a first language and do not understand what the 'face to viral variants' means.

Line 16:  Provide clarity on ''affected the transmissibility'' by saying 20% increase in transmissibility. etc.

Line 19: Explain variability. Do you mean emergence of variants? Evolution?

Line 24-25: Explain / Rephrase. What do you mean by 'major public health burden'?

Line 44: Add 'of'. Triggering of a natural response

Line 46: Explain / Rephrase 'infection area'

Line 49: Explain / Rephrase 'may be unbalanced'

Line 61: Replace 'authorized on the market' with 'authorized for commercial use'

Line 68-69: Strictly speaking, the UK variant was sequenced in UK but emerged in Spain. Please correct.

Line 69: Please specify what is meant by 'high number of mutations'

Line 75: Please explain that vaccination was developed using the Spike protein from the Wuhan strain. Mutations could affect the ability of vaccines to work because antibodies generated by administration of the vaccine might be less effective against the mutated Spike protein, as seen in emerging variants.

Line 77: Add precision. What number are we talking about when we say 'reported and increasing'

Lines 93-96: Long and complicated sentence with unclear messaging. Please rephrase.

Line 133: Please rephrase 'reading errors increase'

Line 135-136: Please rephrase to clarify meaning

Line 140-141: Please rephrase to clarify meaning

Line 149: Replace 'appeared' with 'was sequenced'

Line 150: Can we be more precise on 43-82% more transmissible? It seems to be a big window.

Line 156: Please rephrase to clarify meaning 'this through to cleavage'

Table 1: Correct spelling of 'potential'

Table 1 third row: Replace 'highly transmissible' with 'upto 75% higher transmissibility'

Table 1 Brazil variant: What does 50-120% transmissibility mean? Can we be more precise?

Table 2: Several numbers are listed with an apostrophe instead of a comma. Please replace 15'000 with 15,000 (and other similar numbers)

Table 2 Novavax: In the description, replace 'proteins' with 'Spike trimers' for precision.

Line 551: Please rephrase/explain 'any form of harmlessness'

Line 591: I am not in favour of this argument to develop vaccines in animals. It is impossible to vaccinate 100% animals globally and then which species do you stop at? The authors can present this argument if they feel strongly about it, but will need to add more weight to it by providing evidence. 

Author Response

Dear reviewer,

Thank you for your advices.The new version of the manuscript includes all your suggestions. Please find attached the point-by-point response

Best regards,

Reviewer 3 Report

The manuscript of Chaqroun et al. reviews the state of the art of current SarsCoV2 variants and the impact these variants have on the efficacy of the therapeutics that have been introduced to tackle the pandemic, in particular monoclonal antibodies and vaccines. The manuscript is well written and accessible even for non-experts in the field, and provides detailed tables of variants and therapeutics, which are useful for any further information. Overall, I consider the work worth publishing.

Author Response

Dear reviewer,

Thank you for your opinion. Please find attached the new version of the manuscript, including propositions of others reviewers.

Best regards,

Round 2

Reviewer 2 Report

Thanks for addressing my comments. The manuscript has improved considerably. However, I feel that the language is still not very clear and can be improved. Direct translation from French reflects in the use of some phrases and words that make sense in French but do not translate in the same way in English. Some examples are given below: 

  • use of 'can be occurred' instead of 'can occur' in Line 51
  • use of 'important' instead of 'significant' in Line 73
  • use of 'for the moment' in Line 231
  • use of 'screening tests realized' instead of 'screening tests completed' in Line 336
  • use of 'good recognition' instead of 'effective recognition' in Line 636

I strongly recommend that the manuscript is proof-read by a native English speaker before publication as statements above can impact interpretation.

Author Response

Dear reviewer, thank you for your comments. The manuscript will be checked by the english editing service proposed by MDPI.

Best regards,

Cédric Hartard